# Molecular and Physiological Mechanisms Underlying Submerged Germination in Rice

**DOI:** 10.3390/biology14111470

**Published:** 2025-10-22

**Authors:** Shuang Jia, Qianya Zhou, Shengqi Yuan, Yifeng Wang, Zhongchen Zhang

**Affiliations:** 1College of Agriculture, Northeast Agricultural University, 600 Changjiang Road, Harbin 150030, China; jiashuang9750@163.com (S.J.); 18846223198@163.com (S.Y.); 2State Key Laboratory of Rice Biology and Breeding, China National Rice Research Institute, Hangzhou 311400, China; zqy18173438022@126.com

**Keywords:** *Oryza sativa*, submergence stress, submerged germination, physiological and molecular mechanisms

## Abstract

**Simple Summary:**

Seed dormancy and germination are crucial traits that enable rice to adapt to adverse conditions and maintain its reproductive capacity. Flood (submergence) is a major abiotic stress that inhibits rice seed germination, reducing submerged germination efficiency and seedling uniformity. This review summarizes the molecular and physiological mechanisms of submerged germination in rice, including morphological adaptation, low-oxygen perception, phytohormone interaction, and key genes relevant to breeding flood-tolerant varieties. Emerging technologies, such as whole-genome selection and gene editing technologies, are poised to accelerate the cultivation of rice varieties with enhanced flood tolerance. These advances would provide a theoretical basis and genetic resources to boost crop productivity under hypoxia stress.

**Abstract:**

Submergence during germination (SG) is a major constraint during sowing, severely limiting the promotion and application of direct-seeded rice. Recent studies have revealed the adaptive mechanisms by which rice responds to SG. At the physiological level, flood-tolerant varieties effectively maintain energy supply and cellular homeostasis by enhancing amylase activity, improving glycolysis and ethanolic fermentation efficiency, promoting embryo sheath elongation, and activating antioxidant enzyme systems; at the molecular level, multiple key genes and signalling pathways have been identified, including *SUB1A*, *OsTPP7*, *OsGF14h*, etc., participating in hypoxia perception, metabolic reprogramming, and hormone signal integration to regulate SG under flooding. In addition, the interactions among plant hormones, such as ethylene, gibberellin, abscisic acid, and cytokinin, also play key roles in the SG process. Future research should prioritize breeding strategies that pyramid multiple genes by integrating gene editing, whole-genome selection, and high-throughput phenotyping to improve seed germination under flood stress.

## 1. Introduction

Rice is the primary food crop for over half of the world’s population, and its secure production is essential for a stable food supply. Currently, over 35% of rice paddies worldwide are threatened by floods or inundation, posing a significant threat to food security, particularly in Asia and Africa [1]. With the intensification of climate change, extreme precipitation events are occurring more frequently, and the frequency, intensity, and duration of floods are increasing, making flood stress a primary abiotic stress limiting sustainable rice production [2].

Submerged germination (SG) refers explicitly to the process in which seeds encounter flooding during the germination stage after sowing, resulting in the seeds responding to hypoxia or anaerobic soil conditions, and the germination and seedling formation are severely inhibited [3]. Rice genotypes with SG tolerance can germinate successfully in low oxygen environments through a series of physiological and molecular response mechanisms, including accelerating coleoptile elongation to access surface oxygen, maintaining high amylase activity to mobilize endosperm starch, enhancing glycolysis and ethanolic fermentation pathways to support basic energy (ATP) supply, and regulating ethylene synthesis and the expression of related genes (e.g., Expansins) to promote cell expansion [4].

Under flood stress, flood-tolerant rice varieties efficiently produce ATP through glycolysis and tolerate the accumulation of fermentation products, such as ethanol [5]. The plant hormone network precisely regulates this process: ethylene, a key signalling molecule in low-oxygen responses, promotes seed germination and coleoptile elongation by regulating the biosynthesis and signal transduction of gibberellin (GA) and abscisic acid (ABA) [6]. To date, multiple key genes related to SG have been identified. *OsTPP7* promotes soluble sugar accumulation through the trehalose metabolism pathway, providing energy and osmotic regulation of SG [7]. The CIPK15-SnRK1A signalling pathway activates the expression of glycolysis-related genes under low oxygen conditions [8]; SUB1A not only regulates the flood tolerance of seedlings but also participates in ethylene signal transduction during SG [9]. However, current research still focuses on a few specific significant genes, lacking a systematic understanding of the overall regulatory network of SG.

Therefore, elucidating the molecular network and the multidimensional adaptation mechanism of submerged germination is not only a core scientific question for revealing the evolution of environmental adaptability but also provides a target for breeding crops with flood resistance. In recent years, research has identified key genes, such as SNORKEL/SUB1A in rice, which increases the germination rate to 85%, and EREB180 in maize, which increases the elongation rate of the hypocotyl by 40% under flooding conditions [9,10]. Given the intensified threat of waterlogging, it is of urgent practical significance to deepen the understanding of flood stress response mechanisms and accelerate the utilization of flood-tolerant gene resources for cultivating new varieties with stable production under adverse conditions, thereby ensuring agricultural sustainability and food security. In recent years, studies have provided a comprehensive review of rice’s response to submergence stress throughout its entire growth cycle. These studies systematically summarized key physiological and molecular mechanisms, such as the “quiescence strategy” mediated by the *SUB1A* gene during the vegetative growth stage, the “escape strategy” mediated by the *SNORKEL* gene under deep-water conditions, and the formation of leaf aerenchyma in mature plants, while proposing corresponding integrated agronomic management measures [11,12,13,14]. The research mentioned above primarily focused on the adaptation of rice plants to submergence during both the seedling and mature stages. This review explicitly examines the unique response mechanisms of seed anaerobic germination (AG), particularly by elucidating the perception of hypoxia in the regulatory network governing germination, including morphological adaptation, low-oxygen perception, and hormonal crosstalk, thereby expanding the field of rice submergence stress and providing a crucial theoretical basis and excellent genetic resources for cultivating flood-tolerant rice varieties, especially for direct seeding.

## 2. Physiological Adaptation Mechanism of Submerged Germination

### 2.1. Low Oxygen Stress Response Strategies for Submerged Germination

The low-oxygen environment caused by flooding significantly inhibits seed germination. At this stage, the embryo relies entirely on the breakdown of endosperm storage substances to supply energy, as it lacks an established photosynthetic system. To cope with low oxygen stress, flood-tolerant varieties employ a dual-core mechanism (Figure 1) [15]. First, it involves adjusting carbohydrate metabolism. Under flood stress, embryonic respiratory demand increases, driving efficient starch hydrolysis. The activity of alpha-amylase, the rate-limiting enzyme for starch mobiliation, increases significantly in flood-tolerant varieties, accelerating the conversion of starch to glucose [5]. Consequently, glucose is rapidly produced through glycolysis, and flood-resistant varieties significantly increase the glycolysis ability by enhancing the sensitivity of phosphofructokinase PFK [16]. Second, they activate the fermentation pathways and promote acetic acid recycling. When low oxygen limits the tricarboxylic acid cycle (TAC), ethanol fermentation regenerates NAD^+^ to sustain glycolysis. Flood-tolerant varieties efficiently metabolize acetaldehyde, converting it to acetic acid by aldehyde dehydrogenase (ALDH), and then achieve carbon recovery through acetyl CoA synthase, thereby limiting acetic acid accumulation and providing additional energy [17,18].

Flood stress also triggers the synergistic activation of the seed antioxidant systems (Figure 1). Under flooding, reactive oxygen species (ROS) accumulate, leading to membrane lipid peroxidation and loss of structural integrity, which sharply increases membrane permeability [19]. As a result, nutrients such as ions, soluble sugars, and amino acids leak out while harmful substances enter, ultimately disrupting cellular metabolism and causing cell death [20,21]. Flood-tolerant seeds establish an efficient and synergistic defence mechanism by rapidly activating enzymatic antioxidant systems, including superoxide dismutase (SOD), peroxidase (POD), and catalase (CAT) [22]. Acting together, these enzymes effectively eliminate ROS levels, limit membrane permeability, and maintain high levels of reduced glutathione (GSH) in flood-tolerant compared with sensitive varieties [23,24].

High GSH levels play multiple key roles: GSH directly scavenges free radicals, enhancing the enzymatic antioxidant defence ability, and it serves as a core cofactor in the ASA-GSH cycle, maintaining antioxidant enzyme system [25]; In addition, GSH neutralizes toxic metabolites through detoxification, and maintains a reducing environment within cells to protect protein function and regulate stress-related genes expression. Ultimately, enzymatic antioxidants and non-enzymatic GSH systems work together to maintain cellular redox homeostasis, significantly enhancing the survival and recovery ability of plants under flood stress [26].

### 2.2. Morphological Adaptation Mechanism of Submerged Germination

Under flood stress, the rice coleoptile elongates rapidly through multiple coordinated regulatory mechanisms (Figure 1) [18]. At the cellular level, coleoptile elongation under flood stress depends on synergistic regulation of hormones and cell wall dynamics: hypoxia upregulates the glycosyltransferase *OsUGT75A*, catalyzing the inactivation of ABA and JA by glycosylation and relieving their inhibitory effects on coleoptile growth, and activates the expansion proteins to drive the longitudinal expansion of the basal thin-walled cells, significantly enhancing cell elongation [27,28].

Under certain water depth conditions, coleoptiles of flood-resistant varieties elongate rapidly and can reach the water surface within a few days. Synchronous development of longitudinal air chambers enables efficient delivery of oxygen to the endosperm [29]. After emergence above the water surface, the oxygen diffusion rate of the basal aeration tissue significantly increases, accelerating the growth of embryonic roots and the activating leaf primordia. Genetic differentiation analysis suggests that the dominant haplotype, which exhibits high expression of *OsUGT75A*, in japonica rice significantly enhances the coleoptile elongation ability compared with indica rice [27].

## 3. Molecular Regulatory Mechanism of Submerged Germination

### 3.1. Low Oxygen Perception During Flooded Germination

The primary challenge during submerged germination is low oxygen stress caused by rapid oxygen depletion in the soil environment. To address this, plants employ a low-oxygen sensing system centred on the ERF-VII transcription factor family, whose stability is precisely regulated by the N-terminal degradation pathway, which is highly dependent on oxygen concentration [11]. Plant cysteine oxidases (PCOs), molecular oxygen sensors, are key components of this pathway [30]. Under normal conditions, the second conserved N-terminal cysteine of ERF-VII protein is specifically oxidized by PCOs to form cysteine sulfinate [31,32,33]. This modification subsequently triggers arginine (Arg) labelling mediated by arginine tRNA transferase, forming a typical N-terminal degrader that is recognized by ubiquitin ligase for degradation, resulting in the inability of ERF-VII to accumulate in the cytoplasm [31,34]. Under low-oxygen conditions, the activity of PCOs is significantly inhibited due to limited molecular oxygen, impairing the oxidative modification of the N-terminal cysteine of ERF-VII. Consequently, subsequent Arg labelling and ubiquitination reactions cannot be initiated, and the proteasome-mediated degradation of ERF-VII is blocked in the cytoplasm [32,34]. Subsequently, with the aid of the nuclear localization signal, the stable ERF-VII protein is transferred into the nucleus and gradually accumulates [33]. In the nucleus, ERF-VII serves as a transcription factor and recruits RNA polymerase II and transcriptional cofactors, binds to the hypoxia-responsive elements in downstream target genes promoters and activates these genes expression, leading to physiological processes such as glycolysis, fermentation metabolism, reactive oxygen species balance, and ethylene signalling response to enhance the survival rate and adaptability of plants in low oxygen environments [35].

The low-oxygen sensing network integrates multiple auxiliary signals to enhance response efficiency [36]. The cytoplasmic calcium ion oscillation induced by flooding is transmitted through the calmodulin and CBL-CIPK systems, synergistically regulating SnRK1 kinase and transcription factors, thereby promoting starch degradation and glycolysis [37,38,39,40]. SnRK1 phosphorylates glycolytic enzymes to enhance glycolysis and is also involved in the ERF-VII pathway [41]. On the contrary, TOR kinase activity is inhibited under low-oxygen conditions, inducing the expression of autophagy-related genes to recover nutrients [41]. Reactive oxygen species (ROS) fluctuations in the early stages of hypoxia are regulated by redox-sensitive proteins [42]. In addition, hypoxia synergizes with ethylene signalling, as hypoxia promotes the expression of ACC synthase, leading to an increase in ethylene. Ethylene, in turn, upregulates ERF-VII expression through the EIN3/EIL1 pathway, forming a positive feedback loop [43,44].

In summary, plants perceive oxygen fluctuations through the PCO-N-degron-ERF-VII core framework and integrate Ca^2+^-SnRK1/TOR-ROS hormone networks to achieve multi-level adaptive responses (Figure 2). This system ensures energy supply and redox homeostasis and promotes gas diffusion through cell wall reconstruction, thereby safeguarding submerged germination in hypoxic soils.

### 3.2. Hormonal Regulation of Submerged Germination

The initiation of rice germination under flood stress depends on fine phytohormone regulation. Salicylic acid (SA) acts as a key specific initiating signal, and its synthesis is induced by flooding [45]. Flooding activates the expression of the key β-oxidation enzyme *OsCNL1/2*, driving conversion of cinnamic acid to SA through the β-oxidation pathway in rice peroxisomes [46]. This process is crucial for submerged germination, as the SA content and germination rate of the *cnl1/2* double mutant significantly decrease under waterlogging; however, exogenous SA restores the germination rate [47]. Moreover, SA activates the auxin aminoconjugatase gene *OsGH3.2/4/8*, which catalyzes the conjugation of active auxin IAA with aspartic acid/alanine to form an inactive form, IAA-Asp/IAA-Ala, and relieves IAA’s inhibition on submerged germination [47].

Flood stress fine-tunes the antagonistic relationship between gibberellin (GA) and abscisic acid (ABA) [48]. The *SD1* gene encodes a rate-limiting enzyme in GA biosynthesis; increased SD1 expression promotes active GA4 production, driving internode elongation and facilitating escape from flooding [49]. By contrast, ABA inhibits germination by maintaining dormancy, while OsUGT75A-mediated ABA glycosylation blocks ABA activity, indirectly enhancing GA effects [27]. The 14-3-3 protein OsGF14h inhibits SnRK2 kinase activity, weakening ABA signalling, while stabilizing SD1 to promote GA biosynthesis under flood stress [50,51].

The interaction between ethylene and GA enables rice to flexibly adjust its growth strategies according to flooding intensity, which is directly influenced by ambient the ethylene concentration [52,53]. Under high-intensity flooding, ethylene accumulates due to limited physical diffusion. And high concentration ethylene activates the OsEIL1a activity, which inhibits GA biosynthesis and enhances GA degradation; In parallel, ethylene stabilize DELLA protein, jointly suppressing meristematic tissue-related gene expression. Together, these actions inhibit internode elongation, reduce energy consumption, and shift plants toward an energy-saving, flood-tolerant mode. On the contrary, under low-intensity flooding, ethylene diffuses effectively and remains below the threshold to activate the inhibitory pathways. GA biosynthesis proceeds and promotes SLR1 degradation through GA signalling, relieving the transcriptional inhibition effect of SLR1 on *SK1/2* and activating the expression of cell cycle proteins and cell wall relaxases, ultimately promoting rapid internode elongation to achieve the strategy of escaping flooding.

Submerged germination also depends on the balance between ABA and cytokinin (CTK) [54]. Abscisic acid (ABA) inhibits germination by enhancing seed coat mechanical resistance, suppressing embryonic root-related genes (e.g., *OsEXPB5)*, and activating the *ABI3/4/5* transcriptional network in rice [55]. Cytokinin (CTK) induces the ABA-degrading enzyme CYP707A and inhibits NCED-mediated ABA biosynthesis. Cytokinin (CTK) also activates type B RRs through the AHK receptor, directly inhibiting ABI5 activity and promoting its degradation. The CTK/ABA ratio in seeds that successfully germinate under flooding is significantly higher than that in dormant seeds. Once this ratio exceeds a threshold, repression of cell wall hydrolases and energy metabolism-related genes is relieved, enabling embryonic roots to penetrate the seed coat. Overall, the hormone network, from SA-specific initiation to the integration of GA/ABA/ETH antagonistic signals, ultimately regulates the adaptive germination of rice under flooding stress through the ABA/CTK equilibrium point (Figure 3).

## 4. Exploration and Molecular Breeding Application of Key Genes for Submerged Germination

To date, multiple key genes that regulate crop submerged germination have been identified (Table 1). For example, *SUB1A* exhibits natural variation related to submerged germination: the flood-tolerant allele SUB1A-1 precisely regulates downstream gene networks by activating ethylene response factors and GA inhibition modules, inducing plants to enter a low-metabolism ‘resting strategy’ state, reducing energy consumption, and enhancing hypoxia tolerance [56]. Field experiments show that introgressing SUB1A-1 into the IR64 background significantly increases seedling survival after flooding [57]. Additionally, OsTPP7 mediates trehalose-6-phosphate metabolism, facilitating rapid elongation of the coleoptile and aiding seedlings in breaking through the water surface [58]. The low-oxygen energy-sensing hub CIPK15-SnRK1A activates anaerobic metabolic-related enzymes, such as ADH and PDC, through phosphorylation, thereby maintaining ATP levels [59]. In addition, several other important flood-tolerant genes have been reported in recent years. For example, maize *ZmEREB180* enhances anaerobic response regulation and improves root survival in low oxygen environments [10]; rice *SNORKEL1* and *SNORKEL2* mediate deepwater adaptive responses, promoting rapid stem elongation through GA signalling to cope with flood stress [60], and HRE1 and HRE2 serve as hypoxia-responsive factors involved in regulating the expression of a series of anaerobic-related genes in *Arabidopsis* [61]. These findings deepen molecular understanding of flood tolerance mechanisms and identify potential targets for multi-gene pyramiding to enhance tolerance during submerged germination.

When applying the above-mentioned key genes to flood-tolerant rice breeding, especially for the critical process of waterlogging germination, precise matching and combination must be carried out based on their functional characteristics and action period. Taking the *SUB1A* gene as an example, its “stationary strategy” has been successfully introduced into multiple main cultivars through molecular marker-assisted selection and promoted in numerous Asian countries [56]. However, there are still significant limitations to directly using *SUB1A* to address flood stress during germination, as its primary function is to promote the rapid recovery of plant growth by utilizing stored carbohydrates after water withdrawal, rather than directly regulating the germination process of seeds in anaerobic environments [56]. Therefore, relying solely on the introduction of the *SUB1A* gene is difficult to effectively solve the core problem of the low emergence rate of rice under flood conditions [62]. In contrast, genes such as *OsTPP7* and *OsEXP4* play a more direct role in promoting rapid elongation of the embryo sheath and assisting seedlings in breaking through the water layer, making them key targets for improving submerged germination [18,58,63]. To achieve complete cycle flood tolerance from germination to seedling growth, breeding strategies need to aggregate *SUB1A* with genes that specifically regulate anaerobic germination. Another gene with great potential for application is *OsGF14h*, which encodes the 14-3-3 protein, serving as a signalling hub to enhance flood tolerance by integrating ABA and GA signalling pathways [40]. The advantage of this gene lies in its ability to coordinate multiple physiological processes upstream, which may endow plants with a more balanced ability to adapt to stress [40]. However, its application still faces bottlenecks, such as genetic background dependence and potential trade-offs in growth. As a key node in the signal network, the effect of *OsGF14h* may vary depending on the genetic background of the variety, posing a challenge to its stable expression in different germplasm resources [64,65]. More importantly, regulating hormone balance to enhance stress tolerance may have unexpected effects on other important agronomic traits, such as excessive plant growth or decreased yield under non-stress conditions [66]. Therefore, when utilizing *OsGF14h*, it may be necessary to adopt precise regulatory strategies, such as promoter engineering, to achieve its specific expression under flood stress, thereby enhancing flood tolerance while minimizing adverse effects on normal growth and development.

Recently, gene editing technology has accelerated the application of key genes regulating submerged germination in the genetic improvement of crop flood tolerance traits [67,68]. On the one hand, using CRISPR/Cas9 technology to knock out or functionally weaken genes that negatively regulate submerged germination effectively enhances crop flood tolerance. For example, knocking out the negative regulatory gene *ZmEREB179* in maize significantly improves the plant’s tolerance to flood stress, with mutants exhibiting higher survival rates under waterlogging [69]. On the other hand, targeted editing of regulatory sequences in key flood-tolerant genes can achieve spatiotemporal specificity or stress-induced expression, thereby enhancing flood stress tolerance. For instance, the low oxygen adaptation-related gene *OsGF14h* cloned from weed rice encodes a type of 14-3-3 protein, and enhancing its allele function by gene editing technology could promote low oxygen signal transduction and regulate the balance between ABA and GA, significantly improving rice germination and seedling formation under low oxygen conditions [50].

**Table 1 biology-14-01470-t001:** Related genes regulate flood stress tolerance in plants.

Category	Genes	Species	Functional Mechanism	References
Energy metabolism and antioxidant activity	*CIPK15-SnRK1A*	rice	Low oxygen energy sensing hub, phosphorylation activates anaerobic metabolic enzymes such as ADH and PDC	[59]
	*OsTPP7*	rice	Mediate trehalose-6-phosphate metabolism	[58]
Hormone related	*SUB1A-1*	rice	Activate the ethylene response factor and gibberellin inhibition module to induce a low metabolism ‘quiescent strategy’	[56,57]
	*SNORKEL1/2*	rice	Mediate deepwater adaptive response and promote stem elongation through gibberellin signalling	[60]
	*OsGF14h*	rice	Encodes 14-3-3 protein, improving hypoxia signal transduction, regulating ABA and GA balance	[50]
	*SD1 (OsGA20ox2)*	rice	Key enzyme for gibberellin synthesis mediates the ethylene gibberellin cascade and promotes rapid stem elongation	[70,71]
Morphological adaptation	*OsEXP4*	rice	Encode expansion proteins, promote cell wall relaxation, and drive rapid coleoptile elongation	[18,63]
Low oxygen perception	*HRE1/2*	Arabidopsis	A hypoxia-responsive factor regulates the expression of anaerobic-related genes	[61]
	*ZmEREB180*	Maize	Enhance anaerobic response regulation	[10]
	*ZmEREB179*	Maize	(Negative regulator) Mutants enhance flood tolerance	[69]

## 5. Future Perspectives

However, this field still faces three major challenges: First, the expression of flood-tolerant genes is susceptible to genetic background interference, leading to insufficient phenotypic stability [12,56]. Second, there is a genetic trade-off between flood tolerance and yield, with most flood-tolerant germplasms showing reduced yields in non-stress environments. Third, the response to flooding stress involves a dynamic and complex gene regulatory network, and its mechanism has not been fully elucidated. The cause of these challenges lies in the limitations inherent in the current research paradigm. On the one hand, the verification of gene function is mainly carried out under highly controlled laboratory conditions, which is significantly different from the complex soil microbial community and dynamic climate environment in the field. The differences between laboratory and field conditions may lead to an overestimation of the effects on gene function, and even result in misleading conclusions, as the phenotypes observed in the laboratory cannot be stably reproduced in complex field environments. On the other hand, genetic loci identified through quantitative trait locus (QTL) mapping and genome-wide association analysis (GWAS) often face significant challenges in transitioning from statistical association to breeding applications due to limited resolution, linkage drag, and substantial genotype-environment interaction effects.

Recently, several emerging technologies have begun to address these bottlenecks. The combination of whole-genome selection and gene editing technology offers a precise solution to overcome the negative genetic trade-off between flood tolerance and yield. For example, based on whole-genome association analysis, a key gene, *qSHS5*, regulating seedling flood tolerance was identified in 322 rice germplasms. The study found that the combination of flood tolerance haplotype qSHS5H3 and semi-dwarf allele SD1H1 synergistically enhances stress tolerance and yield [13]. At the same time, the rice hypoxia tolerance-related genes identified through transcriptome analysis provide new genetic improvement targets for breeding rice flood tolerance. Finally, high-throughput plant phenotype analysis techniques based on deep learning played a key role in analyzing the dynamic network response to flooding stress. This technology has demonstrated significant potential in enhancing stress resistance under various conditions, and its effectiveness has been validated through studies on drought resistance mechanisms. For example, the RPT technology developed by Huazhong Agricultural University, based on the optimized SegFormer algorithm, has achieved dynamic and non-destructive extraction of rice root phenotypes, accurately quantifying the dynamic changes in key phenotypic parameters such as root three-dimensional configuration, biomass accumulation, and allocation. It has been successfully applied to the analysis of drought resistance-related traits in 219 rice populations, identifying key genes, such as *OsIAA8* [72]. Hence, plant phenotype analysis techniques could provide a reliable and efficient technical platform for exploring key genes and their mechanisms that regulate submerged germination.

Looking ahead, future research on the regulatory mechanisms of rice submerged germination could be extended to the following two cutting-edge directions: (I) Epigenetic regulatory mechanisms, aimed at elucidating how flood stress dynamically regulates the expression of flood tolerance-related genes through DNA methylation, histone modification, and other methods [62,73]. (II) The function of seed microbiome and the exploration of how seed probiotic community enhances plant low oxygen tolerance during submerged germination [74]. The exploration of these fields is expected to reveal the epigenetic and environmental interaction regulatory network of submerged germination, providing new approaches for designing and breeding flood-tolerant rice varieties.

## 6. Conclusions

This review focuses on the specific physiological process of rice submerged germination and systematically reviews the topic at three levels: physiological adaptation, molecular regulation, and breeding strategies. It forms an effective complement and deepens previous flood-tolerant reviews, covering the entire growth period. On the other hand, this review reveals the core response axis of “low oxygen perception hormone reprogramming coleoptile elongation “unique to seeds during germination, summarizes key regulatory modules specific to germination, such as SA-IAA signal inactivation, OsUGT75A-mediated ABA glycosylation, and ABA/CTK dynamic balance, and thus constructs a more refined anaerobic germination regulatory network framework. At the application level, molecular marker-assisted selection or gene editing biotechnology could be used to introduce excellent flood-tolerant germination alleles into the main rice varieties, cultivate new germplasm with flood-tolerant germination ability, and ultimately improve energy utilization efficiency, antioxidant capacity, and seedling emergence uniformity under flood conditions, thereby ensuring yield stability.

## Figures and Tables

**Figure 1 biology-14-01470-f001:**
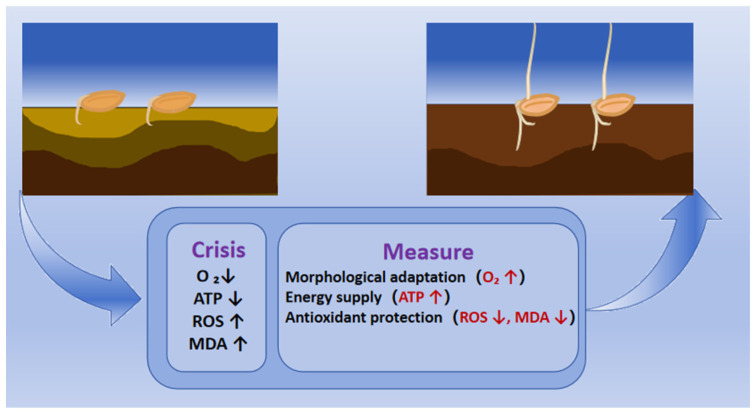
The physiological mechanisms of seed submerged germination. The hypoxic environment (O_2_↓) caused by waterlogging is a critical initial event, which inhibits aerobic respiration, leading to an energy crisis (ATP↓), a burst of reactive oxygen species (ROS↑), and membrane lipid peroxidation (MDA↑). To cope with these multiple stresses, seeds have evolved a coordinated adaptive strategy: morphological construction to facilitate oxygen transport (O_2_↑), metabolic conversion to maintain energy supply (ATP↑), and activation of antioxidant defences to mitigate oxidative damage (ROS↓, MDA↓). The synergistic integration of these processes serves as the physiological foundation for seeds to overcome flood adversity and achieve successful germination. The figure was charted using BioRender https://www.biorender.com/ (accessed on 18 August 2025) according to the website’s instructions.

**Figure 2 biology-14-01470-f002:**
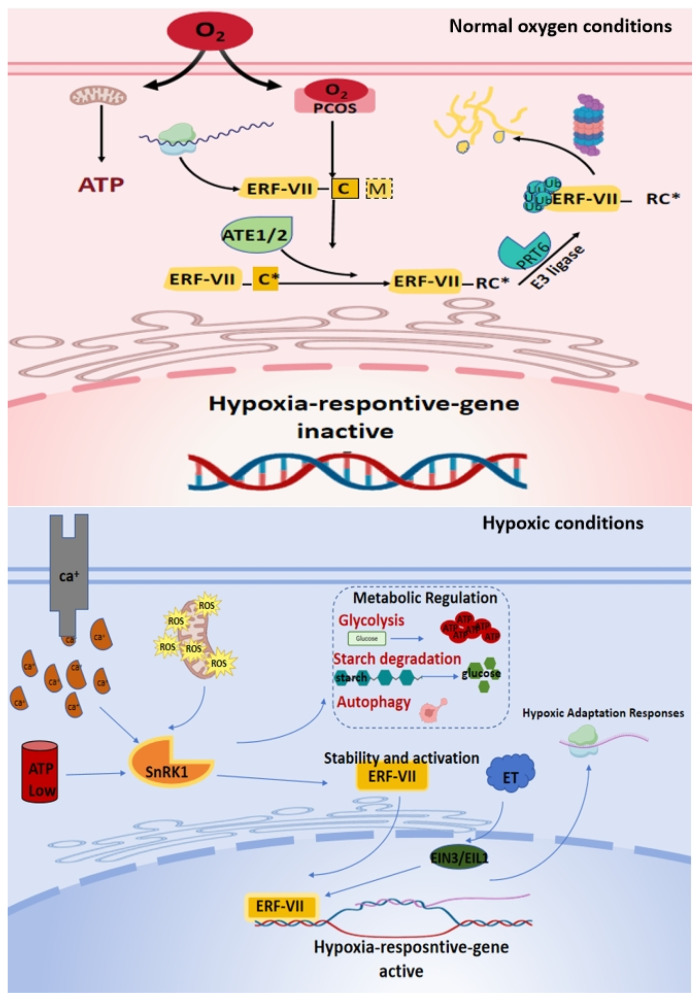
Under normal oxygen conditions (**upper panel**), the ERF-VII transcription factor in plant cells is labelled and degraded by ubiquitination mediated by ATE1/2 and E3 ligases, resulting in the silencing of hypoxia-responsive genes and maintaining normal respiratory metabolism in the cells. C* represents oxidized cysteine, RC* represents arginine-bound oxidized cysteine. Under hypoxic conditions (**lower panel**), hypoxic stress triggers intracellular calcium signalling, the accumulation of reactive oxygen species (ROS), and energy depletion (decreased ATP levels), thereby activating the core regulatory factor, SnRK1 kinase. Activated SnRK1 coordinates adaptive response through dual pathways: (I) initiating metabolic processes such as glycolysis, starch degradation, and autophagy to generate energy rapidly; (II) interacted with the ethylene (ET) signalling pathway activated by hypoxia, the transcription factor ERF-VII is activated, which in turn activates the expression of downstream hypoxia-responsive genes, ultimately promoting cell adaptation to hypoxia stress. The solid arrow represents the promoting effect. The figure was charted using BioRender https://www.biorender.com/ (accessed on 18 August 2025) according to the website’s instructions.

**Figure 3 biology-14-01470-f003:**
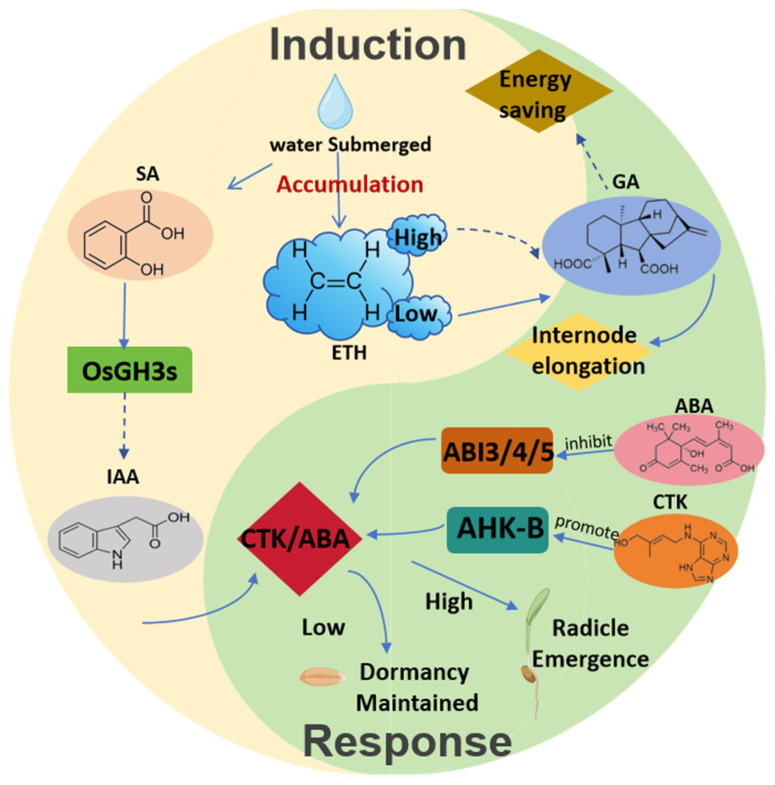
Plants form adaptive response mechanisms through the synergistic action of multiple hormones under flooded and hypoxic conditions. In the initial stage of flooding response, salicylic acid (SA) induces the synthesis and activation of the *OsGH3s* gene, thereby relieving the inhibitory effect of auxin on germination. At the same time, ethylene (ETH) accumulates rapidly, and its concentration directly determines the subsequent response strategy. After entering the response phase, plants exhibit differentiated regulation based on ethylene concentration: high concentrations of ethylene inhibit growth and activate energy-saving tolerance strategies, whereas low concentrations of ethylene promote gibberellin (GA) synthesis, driving internode elongation and executing evasion strategies. At the level of germination regulation, the dynamic balance between cytokinin (CTK) and abscisic acid (ABA) plays a key role: CTK initiates dormancy through AHK-B receptors, while ABA maintains dormancy by regulating the expression of *ABI3/ABI4/ABI5*. The dashed arrows indicate inhibitory effects, while the solid arrows indicate promoting effects. The figure was charted using BioRender https://www.biorender.com/ (accessed on 18 August 2025) according to the website instructions.

## Data Availability

No new data were created or analyzed in this study.

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
