# Peer review of "Molecular and Physiological Mechanisms Underlying Submerged Germination in Rice"

_biology, 2025, doi:10.3390/biology14111470_

Round 1
Reviewer 1 Report
Comments and Suggestions for Authors
This article addresses an important issue: submerged germination of rice. This problem is of global significance and cannot be ignored by scientists, especially in areas where rice is of primary importance. The authors have clearly identified the focus of their efforts. However, we felt that the article insufficiently covered the experience of other studies. For example, a similar topic is discussed in the work of Kumar et al. (2021) https://doi.org/10.1016/j.envexpbot.2021.104448 . However, we found no references to this work in the manuscript under review.
In Chapter 2, "Physiological Adaptation Mechanism of Submerged Germination," we recommend including illustrations. Without them, the information is difficult to digest. The reader would benefit from photographs related to the topic "Morphological Adaptation Mechanism of Submerged Germination."
Chapter 4 contains useful information on the genetics of the problem under discussion, but it could also be expanded.
Therefore, we believe the article should be supplemented and expanded.
Author Response
Comments 1: This article addresses an important issue: submerged germination of rice. This problem is of global significance and cannot be ignored by scientists, especially in areas where rice is of primary importance. The authors have clearly identified the focus of their efforts. However, we felt that the article insufficiently covered the experience of other studies. For example, a similar topic is discussed in the work of Kumar et al. (2021) https://doi.org/10.1016/j.envexpbot.2021.104448 . However, we found no references to this work in the manuscript under review.
Response 1: Thank you for your comments. Revised as required. See lines 80-86 in the revised version.
Comments 2: In Chapter 2, "Physiological Adaptation Mechanism of Submerged Germination," we recommend including illustrations. Without them, the information is difficult to digest. The reader would benefit from photographs related to the topic "Morphological Adaptation Mechanism of Submerged Germination."
Response 2: Nice suggestion. We have added an illustration as required. See Fig.1, lines 164-173 in the revised version.
Comments 3: Chapter 4 contains useful information on the genetics of the problem under discussion, but it could also be expanded.
Response 3: Revised as required. See lines 372-403 and Table 1 in the revised version. The authors appreciate your efforts for our paper, which we believe will significantly improve our article.
Reviewer 2 Report
Comments and Suggestions for Authors
dear authors,
your manuscript is well prepared and reflect a good effort. good jop

Author Response
Comments 1: Line 171, “ROS fluctuations in the early stages of hypoxia are regulated by...”, line 232, “ABA inhibits germination by enhancing seed...”, line 234-235, “CTK induces the ABA-degrading enzyme...” and “CTK also activate type B RRs through ...”, do not start the sentence with abbriviation.
Response 1: Thank you for your suggestions. Corrected as required. See lines 207, 307, 309 and 311 in the revised version.
Reviewer 3 Report
Comments and Suggestions for Authors
The manuscript provides a detailed and well-structured overview of the molecular and physiological mechanisms underlying submerged germination (SG) in rice. However, while the review is comprehensive in terms of gene-level and hormonal regulation, it lacks a critical comparative analysis that situates these findings within the broader context of crop stress physiology and applied breeding strategies. Specifically:
While the article deeply covers signaling pathways and individual genes (e.g., SUB1A, OsTPP7, OsGF14h), it provides limited discussion on how these findings have been or can be translated into breeding programs, field management, or agronomic practices for flood-prone regions. A more critical evaluation of the bottlenecks in deploying these genetic resources (e.g., yield trade-offs, genetic background dependency, or regulatory hurdles for genome editing) would make the review more impactful.
Figures 1 and 2 illustrate molecular and hormonal mechanisms, but they are highly detailed and dense. Simplified schematic models showing (i) the central regulatory hubs, (ii) the crosstalk between hormones, and (iii) the breeding application pipeline (from gene discovery to cultivar release) would improve accessibility for a broader readership.
The “Future Perspectives” section is promising but remains somewhat descriptive. A stronger critical analysis is needed, such as discussing whether multi-gene stacking through CRISPR/Cas or genomic selection is feasible in the near term, or what phenotyping challenges remain unsolved. Highlighting gaps (e.g., epigenetic regulation, seed microbiome roles in SG, or interaction with climate change–induced stresses) could make the outlook more visionary.
Overall, the review is scientifically sound and clearly written, but to maximize its value, it should move beyond compiling existing studies toward a more critical, comparative, and application-oriented synthesis
Author Response
Comments 1: While the article deeply covers signaling pathways and individual genes (e.g., SUB1A, OsTPP7, OsGF14h), it provides limited discussion on how these findings have been or can be translated into breeding programs, field management, or agronomic practices for flood-prone regions. A more critical evaluation of the bottlenecks in deploying these genetic resources (e.g., yield trade-offs, genetic background dependency, or regulatory hurdles for genome editing) would make the review more impactful.
Response 1: Thank you for your comments. Revised as required. See lines 372-403 in the revised version.
Comments 2: Figures 1 and 2 illustrate molecular and hormonal mechanisms, but they are highly detailed and dense. Simplified schematic models showing (i) the central regulatory hubs, (ii) the crosstalk between hormones, and (iii) the breeding application pipeline (from gene discovery to cultivar release) would improve accessibility for a broader readership.
Response 2: We have renewed Figures 1 and 2 as required. See Fig. 2 and 3, lines 259-271, 336-349 in the revised version.
Comments 3: The“Future Perspectives”section is promising but remains somewhat descriptive. A stronger critical analysis is needed, such as discussing whether multi-gene stacking through CRISPR/Cas or genomic selection is feasible in the near term, or what phenotyping challenges remain unsolved. Highlighting gaps (e.g., epigenetic regulation, seed microbiome roles in SG, or interaction with climate change–induced stresses) could make the outlook more visionary.
Response 3: Revised as required. See lines 442-453, 477-486 in the revised version. The authors greatly appreciate your suggestive comments on this manuscript.
Reviewer 4 Report
Comments and Suggestions for Authors
I would like to congratulate the authors on a well-organized and readable work. I appreciate the opportunity to review this article. The article addresses an important topic in rice germination. The topic and results are interesting and appear to contribute to the advancement of science. The title is clear and reflects the content of the article.
This work contains some technical issues that will need to be resolved before the manuscript is ready for publication.
Could the Simple Summary be more concise?
The Keywords should be modified so that they are not the same as the title. The scientific name of the species under study should come first.
Lines 71–83 (Introduction): The justification for this review is too general. The authors should more clearly highlight the knowledge gap this review addresses and explain how it differs from other recent reviews in the field.
Lines 181 and 245: Methodological details of how the figures were made could be added.
Several paragraphs simply summarize the findings of cited works without critical evaluation. A stronger analytical perspective is needed e.g., discussing methodological limitations, conflicting results among studies, and identifying consensus vs. controversies.
Table 1: The table lacks clear categorization. Consider reorganizing by subtopics or mechanisms to improve clarity.
References
Lines 350–400: The reference list is extensive but uneven. Some key recent studies (2022–2024) appear to be missing, while older and less relevant citations are overrepresented. A systematic approach to literature inclusion should be clarified in the Methods section (or at least in the Introduction).
The manuscript requires substantial English editing. Issues include:
Long, complex sentences that obscure the main message.
Repetitive expressions that reduce conciseness.
Author Response
Comments 1: Could the Simple Summary be more concise?
Response 1: Thank you for your comments. Revised as required. See lines 11-20 in the revised version.
Comments 2: The Keywords should be modified so that they are not the same as the title. The scientific name of the species under study should come first.
Response 2: Corrected as required. See lines 36-37 in the revised version.
Comments 3: Lines 71–83 (Introduction): The justification for this review is too general. The authors should more clearly highlight the knowledge gap this review addresses and explain how it differs from other recent reviews in the field.
Response 3: Nice suggestion. We highlight the differences in our review compared to other recent reviews in the field. See lines 80-94 in the revised version.
Comments 4: Lines 181 and 245: Methodological details of how the figures were made could be added.
Response 4: revised as required. See the Figure legend in the revised version.
Comments 5: Several paragraphs simply summarize the findings of cited works without critical evaluation. A stronger analytical perspective is needed e.g., discussing methodological limitations, conflicting results among studies, and identifying consensus vs. controversies.
Response 5: revised as required. See lines 422-436, 442-453 in the revised version.
Comments 6: Table 1: The table lacks clear categorization. Consider reorganizing by subtopics or mechanisms to improve clarity.
Response 6: Nice suggestion. Revised as required. See Table 1 in the revised version.
Comments 7: The manuscript requires substantial English editing. Issues include: Long, complex sentences that obscure the main message.
Response 7: revised as required. This manuscript was critically edited by Prof. Furong Liu (Peking University), who has studied in the USA for over ten years and is well-versed in English grammar and expression.
Comments 8: Lines 350–400: The reference list is extensive but uneven. Some key recent studies (2022–2024) appear to be missing, while older and less relevant citations are overrepresented. A systematic approach to literature inclusion should be clarified in the Methods section (or at least in the Introduction).
Response 8: we have added some key recent studies (2022-2024) as required, particularly in Chapter 4 of the revised version, and most of the references focus on rice submerged seed germination in response to flood stress. The authors greatly appreciate your suggestive comments on this manuscript.
Round 2
Reviewer 1 Report
Comments and Suggestions for Authors
The authors have made the appropriate changes we previously recommended. We have decided that this manuscript can now be published.
Reviewer 3 Report
Comments and Suggestions for Authors
The authors have responded to the required comments